# Exploring the use of social media and online methods to engage persons with lived experience and healthcare professionals in creating research agendas: Lessons from a pediatric cancer research priority-setting partnership

Kyobin Hwang[1‡], Surabhi Sivaratnam[1,2‡], Rita Azeredo[1], Elham Hashemi[1], Lindsay A. Jibb[1,2]*

**1** Hospital for Sick Children, Toronto, Canada, **2** University of Toronto, Toronto, Canada

‡ These authors share first authorship on this work.
* lindsay.jibb@utoronto.ca

**Data Availability Statement:** All relevant data are within the manuscript.

## Abstract

Social media is increasingly used to engage persons with lived experience and healthcare professionals in research, however, there remains sparse guidance on how to effectively use social media to engage these groups in research agenda-setting. Here we report our process and experience utilizing a social media campaign to engage Canadians within the pediatric cancer community in a research priority-setting exercise. Following the James Lind Alliance method, we launched a priority-setting partnership (PSP) to develop a child with cancer-, survivor-, family member-, and healthcare professional-based Canadian pediatric cancer research agenda. Social media-based strategies were implemented to recruit participants for two PSP surveys, including preparatory activities, developing a website, launching graphics and advertisements, and engaging internal and external networks. Descriptive statistics of our data and analytics provided by the platforms are used presently to report our process. The framework we implemented involved preparing for social media use, identifying a target audience, developing campaign content, conducting the campaign, refining the campaign as needed, and evaluating its success. Our process resulted in a substantial social media-based reach, good survey completion rates, and a successfully developed pediatric cancer community-specified research agenda. Social media may represent a useful approach to engage persons with lived experience and healthcare professionals in research agenda development. Based on our experience, we present strategies to increase social media campaign engagement that may be useful to those seeking to conduct health research priority-setting exercises.

**Funding:** The study was supported by the CIHR Catalyst Grant in Patient-Oriented Research (#PAO-169422 to LJ). The funders had no role in study design, data collection and analysis, decision to publish, or preparation of the manuscript.

## Author summary

Little is known about how best to use social media to engage people with lived experience and healthcare professionals in research agenda-setting. We present our process and experience using a social media campaign to engage Canadians within the pediatric cancer community in a building such an agenda. We used social media and a network of partners to recruit participant into two research agenda building surveys. The framework we implemented involved preparing for social media use, identifying a target audience, developing campaign content, conducting the campaign, refining the campaign as needed, and evaluating its success. Our process resulted in a substantial social media-based reach, good survey completion rates, and a successfully developed pediatric cancer community-specified research agenda. Based on our experience, we present strategies to increase social media campaign engagement that may be useful to those seeking to conduct health research priority-setting exercises.

## Background

In recent years social media has gained an important role in healthcare, including engaging persons with lived experience and healthcare professionals in research [1,2]. Researchers are increasingly utilizing platforms including Facebook, Twitter, and YouTube to enable research participant recruitment [3–5] and to disseminate study findings [6]. Social media-based methods have also been used to enable the engagement of individuals with lived healthcare experiences (i.e., patients, family members, clinicians, and other advocates) in setting research priority agendas, though there remains sparse methodological guidance on how to do so [7]. The development of such agendas is critically important to direct clinical practice and policy decisions but, to date, those with lived experience have not been routinely involved in the process [8–10].

Considering the imperative to engage these individuals in research agenda building and the potential opportunities to support such work through online methods, our research team utilized social media, amongst other online tools, to build our pan-Canadian pediatric cancer research priority setting partnership (PSP). In brief, through our PSP, we surveyed the Canadian childhood cancer community to elicit their research questions, and subsequently engaged the group in a priority setting activity to develop a childhood cancer research agenda. To create an inclusive agenda, we aimed to engage a bilingual (English and French) and diverse group of children with cancer, pediatric cancer survivors, as well as their family members and healthcare professionals.

Here we review a specific case where social media and other online modalities were leveraged to engage Canadians within the pediatric cancer community in a research priority-setting exercise. As a resource for other research teams, we offer descriptions of our social media-based recruitment strategy, engagement with our social media campaign, and factors associated with increased involvement in our PSP. We also discuss the limitations of our efforts and make specific recommendations on how other research teams might use social media to involve persons with lived experience and healthcare professionals in research agenda-setting.

## Methods

### Overview of pediatric priority setting partnership

We followed the James Lind Alliance process to develop a person with lived experience and healthcare professional-engaged research agenda [11]. At the onset of our PSP, a steering group, comprised of childhood cancer survivors (n = 5), family caregivers (n = 4), and healthcare professionals involved in pediatric oncology (n = 6). The steering group guided all

methodological and operational decisions for our PSP. We launched a national bilingual (French and English) online survey in Winter 2020 to collect research questions from the Canadian pediatric cancer community. Collected questions were collapsed and collated into summary questions, which were checked against the extant scientific literature. Adequately addressed questions were removed from the question roster and a second national, bilingual online survey was launched in Winter 2021 requesting children with cancer, survivors, family members, and health professionals prioritize the unanswered research questions. A shortlist of these questions was then taken to a pan-Canadian consensus-building workshop conducted in March 2022, where the top ten research priorities in Canadian pediatric oncology were established. Social media and other online tools were utilized to promote both of our online surveys. Ethics approval for this study was received from the SickKids Research Ethics Board (REB). Consent for publication was not applicable

## Social media campaign overview

The social media campaign used to promote participation in both online PSP surveys was spearheaded by personnel on our team with experience in digital marketing and graphic design (SS). A trained graphic designer with assisted with designing the graphics for our posts, while another member of the team, who is also a childhood cancer survivor (RA), received training and supported the process by maintaining an active presence across our platforms during the campaigning period (e.g., regularly posting, engaging with comments on posts, etc.). A multi-phase social media campaigning approach was implemented, closely mirroring the framework developed by Lang et al. for recruiting participants in pediatric research through social media [12]. Specifically, we: (i) planned for social media use as an engagement strategy, (ii) identified and attempted to understand the different target audiences and then developed a strategy accordingly, (iii) developed and designed campaign content, (iv) implemented, monitored and iteratively refined campaign strategies, and (v) evaluated campaign success.

## Establishing analytic tools

Dedicated social media accounts were created on Facebook (www.facebook.com), Twitter (www.twitter.com), Instagram (www.instagram.com), LinkedIn (www.linkedin.com), You-Tube (www.youtube.com), and TikTok (www.tiktok.com). Wherever applicable, we registered our "business/professional" account types. This registration type enabled viewing of social media analytics, which included overviews of the demographic characteristics of those engaging with our social media content (i.e., age, gender, education levels, job titles, location, language) and page insights (i.e., likes, comments, shares, page views, page traffic and activities-including the length of time of individual sessions). Among the various utilized platforms, our social media campaigning efforts concentrated on Facebook, Twitter, and Instagram, as well as our study website as our PSP steering group felt our target participants were mostly likely to use these platforms. We also created a Hootsuite account (Hootsuite Inc), which provides access to aggregated analytics of all user profiles across networks, including the most common times of day during which followers interact with social media accounts/pages. The Hootsuite platform also enabled the research team to schedule posts and provided a single point to launch posts across various social media platforms.

## Social media engagement

**Paid advertisements.**   During the social media campaign, we periodically implemented paid advertisements on Facebook, Instagram, and Twitter. Our team closely monitored ad-

associated analytics to identify ways to augment the campaign as needed. For example, we identified that video posts produced more social media engagement than image-based posts, and all future advertisements were then solely accompanied with videos. Each social media platform required specifications on ad delivery, ad content, design language, targeted audience, including location, and dates of deployment.

**Website.**    Prior to launching our social media accounts, a website for the research project was created (www.pedcancerpsp.ca). This website was created using Wix (Wix.com Ltd). Website design considered accessibility needs by ensuring alternative text was available for all graphics and the website optimized for desktop and mobile viewing to support ease of access over a variety of devices. We used the Wix search engine optimization (SEO) checklist to ensure our site could be found via search phrases on various search engines. The website provided a centralized location for prospective participants to learn more about the project, the James Lind Alliance methodology in general, the research team, and registration to our mailing list.

**Engagement supported by internal and external networks.**    Throughout our social media campaign, we contacted external networks, including Canadian childhood cancer organizations, advocacy groups with prominent social media accounts, and non-governmental organizations affiliated with pediatric cancer, and subsequently requested their assistance in promoting the project. Contact methods included email correspondence, as well as direct messaging via social media platforms. When individuals and organizations agreed to help with survey promotion, we provided them with a template package, which included tailored graphics, pre-written messaging, sample newsletter graphics, and draft emails that they could send to their contact lists. A similar methodology was implemented to support dissemination by our internal network (i.e., members of the research team and members of the PSP steering group).

## Data analysis

Facebook, Instagram, and Twitter provide descriptive data analytics to page administrators, allowing for monitoring and assessment of the usage of a page; a similar tool is available on Wix. These analytics allowed us to measure user engagement within each online platform. For Instagram, Facebook, and Twitter, we monitored reach, impressions, and clicks. For our website, we monitored unique visitors, site sessions, and page views. We also collected demographic information from PSP survey participants. We used descriptive statistics to analyze all data types.

## Results

### (I) Preparatory activities for social medial campaign

**Scoping review.**    Prior to launching our social media campaign, a scoping review was conducted to describe: (i) the existing literature on social media–based strategies used to enhance participation of persons with lived experience and healthcare professionals in health research priority-setting, (ii) recommendations for social media–based research priority-setting campaigns, (iii) the benefits and limitations of the method, and (iv) recommendations for future campaigns [7]. Our review identified a total of 23 papers reporting on 22 unique studies. The central findings of our review allowed us to identify potentially useful social media platforms and other online tools to leverage and social media strategies that might enhance our engagement with people with lived experience of childhood cancer and healthcare professionals. The results of this review were presented to our PSP steering group and guided the social media strategies implemented in this study, including the process of creating our accounts on Facebook, LinkedIn, Instagram, Twitter, and TikTok.

Table 1. Time spent preparing for the social media campaign.

| Task | Description of Task | Time (hours) |
|---|---|---|
| Literature review | • Review of existing literature, including searching scientific databases and grey literature to consolidate known social media campaign strategies | 100 |
| Social media platform creation | • Creation of accounts on Facebook, LinkedIn, Instagram, Twitter, and Tiktok<br>• Creation of brand unified profile picture and profile description | 10–20 |
| Website creation | • Brainstorming with research team and steering group on website layout<br>• Creation of website<br>• Compiling of graphics and pictures for the website<br>• Liaising with communications and public affairs department to ensure website meets organization requirements | 200–250 |
| Researching external organization | • Development of a repository of external Canadian childhood cancer organizations to contact for support in promoting PSP surveys<br>• Contacting and liaising with external organizations with requests for survey promotion | 20–40 |
| Brand template | • Generation of branding package, including standardized fonts, colours, graphics, and design elements, as well as centralized messaging and key phrases | 20–40 |
| Promotional videos | • Brainstorming with research team and steering group on the style and content of two videos designed to explain and promote PSP surveys<br>• Generation of video storyboard and script and creation of videos in both English and French | 49 |
| Sample graphic generation | • Creation of Instagram, Facebook, Twitter, Tiktok posts, including static graphics and videos/gifs<br>• Creation of email flyers for external and internal network<br>• Translation of all graphics and other post content from English to French for bilingual posting | 100–150 |
| Creating promotional package | • Creation of individualized promotional packages for internal and external organizations/individuals to support them in PSP survey promotion. Each promotional package contained sample graphics, as well as associated text that organizations/individuals could use to post on their respective platforms | 40–50 |
| Steering committee meetings | • Person with lived experience and healthcare professional continued engagement through routine steering group meetings where social media campaign plans and progress were discussed | 5–10 |
| **TOTAL:** | | **595–709** |

**Building platform accounts and branding.**    Once we created accounts on Facebook, LinkedIn, Instagram, Twitter, and TikTok, a unified brand identity was developed in coordination with our steering group. The brand identity involved employing consistent fonts, colors, graphics, design elements, and messaging. We then began engagement with our external network of Canadian cancer organizations, including by creating the customized promotional packages for these groups. Cumulative the preparatory phase of the campaign required 525–710 hours of research support time. **Table 1** summarizes additional undertakings during this phase of the social media campaign and associated time allocations for each task.

## (II) Identifying and understanding target audience

To gain a deeper understanding of the interest and online behaviours of children with cancer, childhood cancer survivors, family members of patients and survivors, and the healthcare professionals caring for these groups, we regularly consulted with an expert steering group. Co-design workshops were regularly held to build an understanding of each audience and support the development of tailored graphics. The steering group also assisted in recruiting external partner organizations to our network who assisted in disseminating our social media content.

## (III) Developing and designing campaign content

**Creation of social media content.**   An initial branding package was created in Adobe Photoshop (CC 2017; Adobe, San Jose, CA). During the development of these graphics, the designers utilized WebAim (https://webaim.org/resources/contrastchecker/) to ensure all graphics fulfilled the minimum colour contrast requirements for online and mobile accessibility. This package was again reviewed by the PSP steering group to help ensure the content and design was appropriate for the targeted audience. Steering group suggestions were used to modify and finalize the branding package. This branding scheme was used as the basis for a hub of static graphics and videos that would be utilized in the social media campaign.

**Social media campaign content.**   The content of our social media posts included reviews of PSP goals, the key personnel involved, calls for those with lived experience of childhood cancer and healthcare professionals to complete each survey and, intermittently, general posts considered relevant to the childhood cancer community (e.g., recognizing childhood cancer awareness month). Posts typically included graphics in the form of photos, videos, and graphics interchange format (GIFs), and associated text included hashtags, links to the project website or the survey, tags, and various relevant emojis. Sample graphics are shown in **Figs 1** and **2**. Throughout the social media campaign, we monitored: (i) social media analytics to identify gaps in interaction with the campaign and (ii) demographics PSP self-reported by survey participants to identify gaps in communities of individuals completing each survey. The identification of these gaps subsequently informed the content of future social media posts. For example, during the campaign, these metrics identified a lower-than-expected response rate from individuals from the Canadian province of Quebec and efforts were made to target graphics towards this group. Sample targeted graphics are shown in **Fig 3**. We also monitored social media analytics, which provided data on the optimal timing for posts, including the times at which our audience was generally online and interacting with our posts. Our posting schedule was continuously adjusted to reflect these optimal times.

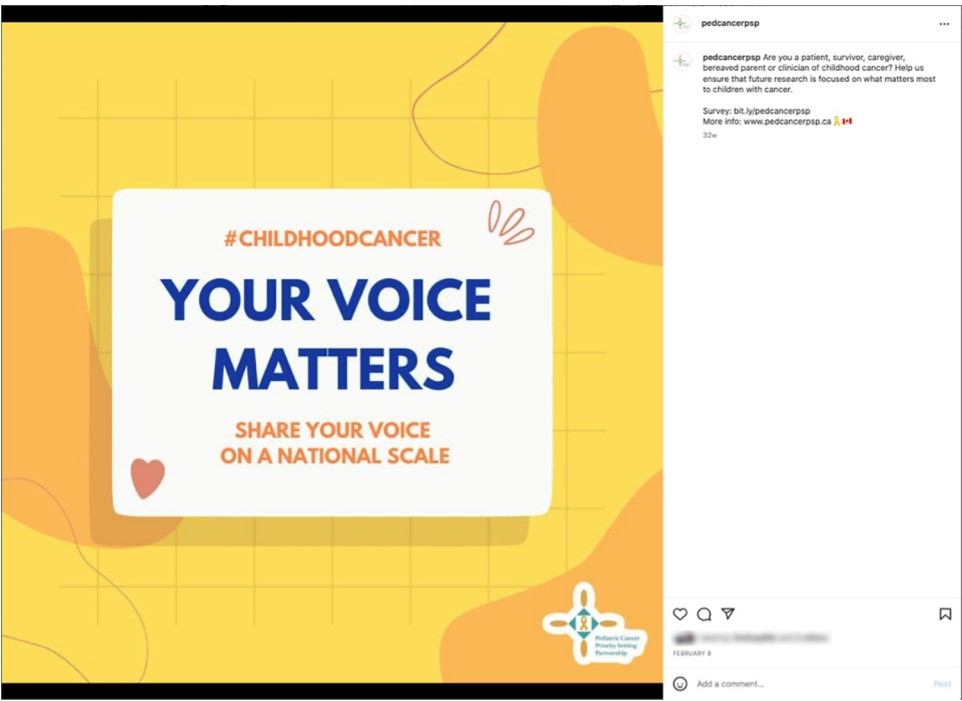

**Fig 1. Example of social media promotional graphic in English.**

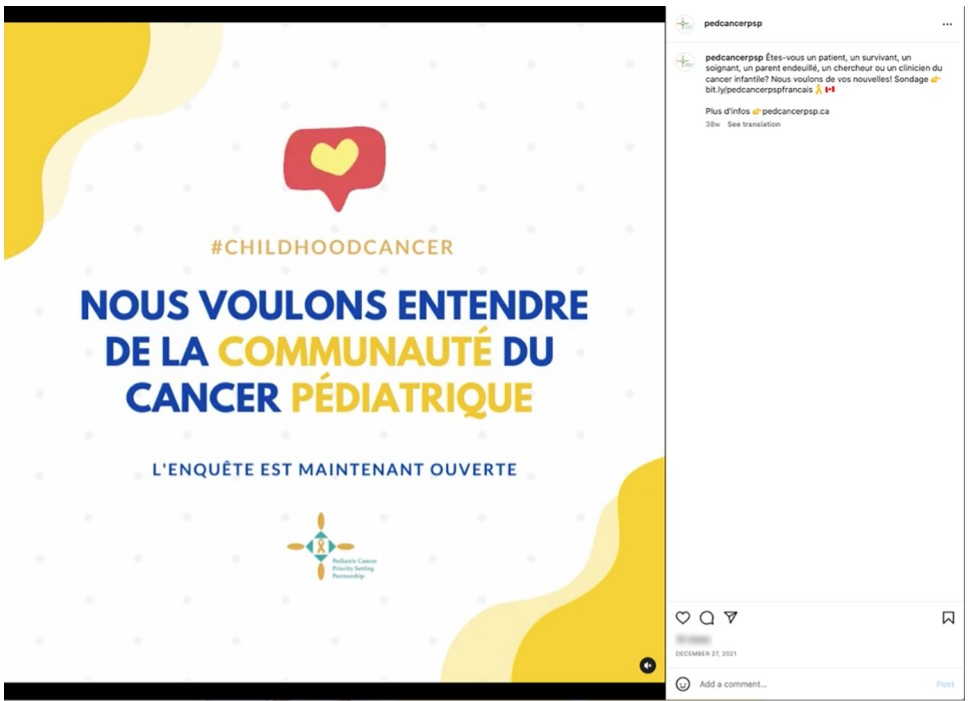

**Fig 2. Example of social media promotional graphic in French.**

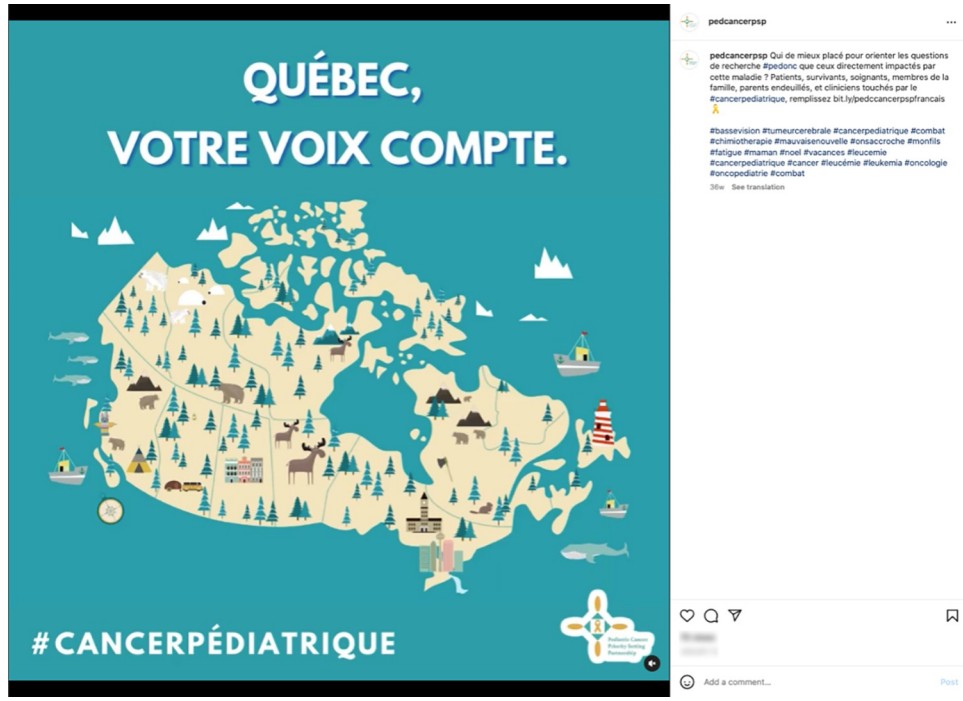

**Fig 3. Example of graphics targeting a specific group.**

## (IV) Implementing, monitoring and iteratively refining campaign strategies

**Initial launch of social media campaign.** A series of introduction posts were generated with the intention of establishing credibility related to our campaign. The first posts identified the research team, and the steering group and subsequent posts introduced the concept of a PSP. At the initial launch of our social media accounts, we contacted established childhood cancer organizations with social media presence through direct messaging to introduce the PSP. We then followed the accounts of these organizations to increase the number of people interacting with our social media content.

**PSP survey participation.** The first survey was available for completion from December 7, 2020, to March 3, 2021. During this period, a total of 330 individuals participated, of which 80 (23.9%) were childhood cancer patients and survivors, 179 (53.4%) were family members of children who have or had cancer, and 76 (22.7%) were healthcare professionals. There was representation from all of Canada's provinces, though there were no response from territory residents and most participants lived in the province of Ontario (n = 160, 47.8%), with the second largest group living in British Columbia (n = 54;16.1%).

The second survey, focused on research question prioritization, was available for completion from January 10, 2022, to February 19, 2022. A total of 197 individuals participated and 49 (24.5%) were children or survivors of childhood cancer, 102 (51.0%) were family members, and 49 (24.5%) were healthcare professionals. Similar to the initial survey, there was representation from all the provinces, though there were no respondents from the territories. Most survey respondents were once again Ontario residents (n = 84, 42.0%), followed by Quebec (n = 59, 29.5%) and Alberta (n = 16, 8%).

**Campaign maintenance.** To support the maintenance and coherence of our social media campaign, a structured set of strategies was implemented. We initiated a graphic generation process that created foundational but modifiable templates for our campaign's visual content informed by our branding scheme. Weekly analysis of platform analytics was pivotal to modifying these templates and reviews of engagement metrics and performance data informed revisions to our content (e.g., if certain content was popular it would direct our next template modifications). The goal of this is iterative process was to maximize campaign engagement.

To ease the process of regularly posting across our platforms, graphics were systematically scheduled for posting on designated days at the start of each week based on or analytics data, ensuring a consistent and predictable content flow. Paid ads were strategically incorporated during select campaign weeks. This necessitated careful analysis of platforms and coordination with internal financial management to optimize the impact of these promotions. The total cost of social media advertisement was $389.91 for both the first and second survey. **Table 2**

**Table 2. Costs of social media advertisements.**

| Platform | Date | Money Spent |
|---|---|---|
| Facebook and Instagram* | 02/03/2021–03/04/2021 | 170.00 |
| Twitter | Feb 27, 2021 | 54.10 |
| Twitter | Feb 25, 2021 | 20.00 |
| Twitter | Mar 2, 2021 | 39.31 |
| Facebook and Instagram* | Jan 27, 2022 | 56.50 |
| Twitter | Jan 19, 2022 | 19.19 |
| Twitter | Jan 28, 2022 | 30.81 |
| **TOTAL:** | | **$389.91** |

*The ad costs for Facebook and Instagram are grouped as they are linked under the same company.

**Table 3. Time spent maintaining the social media campaign on a weekly basis.**

| Platform | Description of Task | # of Hours/ week |
|---|---|---|
| Social media graphics generation* | Creation of a series of base graphics that served as a template for all future graphics. | 4–10 |
| Emails to external organizations | Searches for and liaising with new possible external organizations/ individuals to disseminate social media content and promote PSP surveys | 2–4 |
| Social media post text generation | Generation of text for post descriptions to be reviewed, vetted and then approved for posting by steering committtee | 2 |
| Posting graphics to platforms | Preparation of weekly posting schedule. | 2 |
| Translation | Translation of post graphic and text content from English to French. | 2 |
| Paid advertisements | Creation of and fee payment processing for paid ads | 0–2 |
| Interacting with users | Personalized engagement with target audience | 0–1 |
| Reviewing analytics from the campaign | Weekly review of analytics from each social media platform to optimize posts for subsequent week | 2–5 |
| Brainstorming | Weekly reflection on our campaign and review of posts from other users/organizations to develop ideas to modify posts for subsequent week | 1 |
| **TOTAL:** | | **15–30** |

summarizes cost information. Social media ads were primarily utilized in early campaign periods, but after establishing a substantial and self-sustaining following base, we relied more on sharing content with our existing audience.

Interaction with users was a dynamic aspect of our campaign. We actively engaged with our target audience, addressing questions and responding to comments on posts. This personalized engagement was intended to help foster a sense of community and further amplified our campaign reach. **Table 3** illustrates the hour breakdown and required tasks for the week.

**Strategy modification.** During the campaign, certain approaches were modified or phased out due to practical or strategic considerations. TikTok and LinkedIn platform layouts and post norms created challenges to delivering content that resonated with our particular audience. Thus, these components of our strategies were gradually phased out to more effectively allocate our campaign resources. We also discontinued our initial "following spree" where we proactively followed of pediatric oncology-related accounts across platforms after achieving a substantial and self-sustaining following base, which facilitated our growth. We also phased out the resharing of posts from other users related to pediatric cancer due to the increasing effort required to ensure the credibility of the content shared and associated users. Further, overtime, we shifted from posting separate but identical French and English content to posting bilingual posts that consolidated the messaging for both linguistic communities. Similar considerations were taken into account for video content, where the creation of separate videos for different platforms was streamlined into a unified approach, maximizing our efficiency in content distribution.

## (V) Evaluating campaign success

**Social media campaign summary.** Over the course of our entire PSP—from April 2020 to September 2022—we garnered 870 Instagram followers, 450 Twitter followers, 69 Facebook page likes, 27 TikTok followers, and 20 LinkedIn followers. **Figs 4 and 5** provide an overview of how survey response rates correspond to key timepoints and efforts during the social media campaign period.

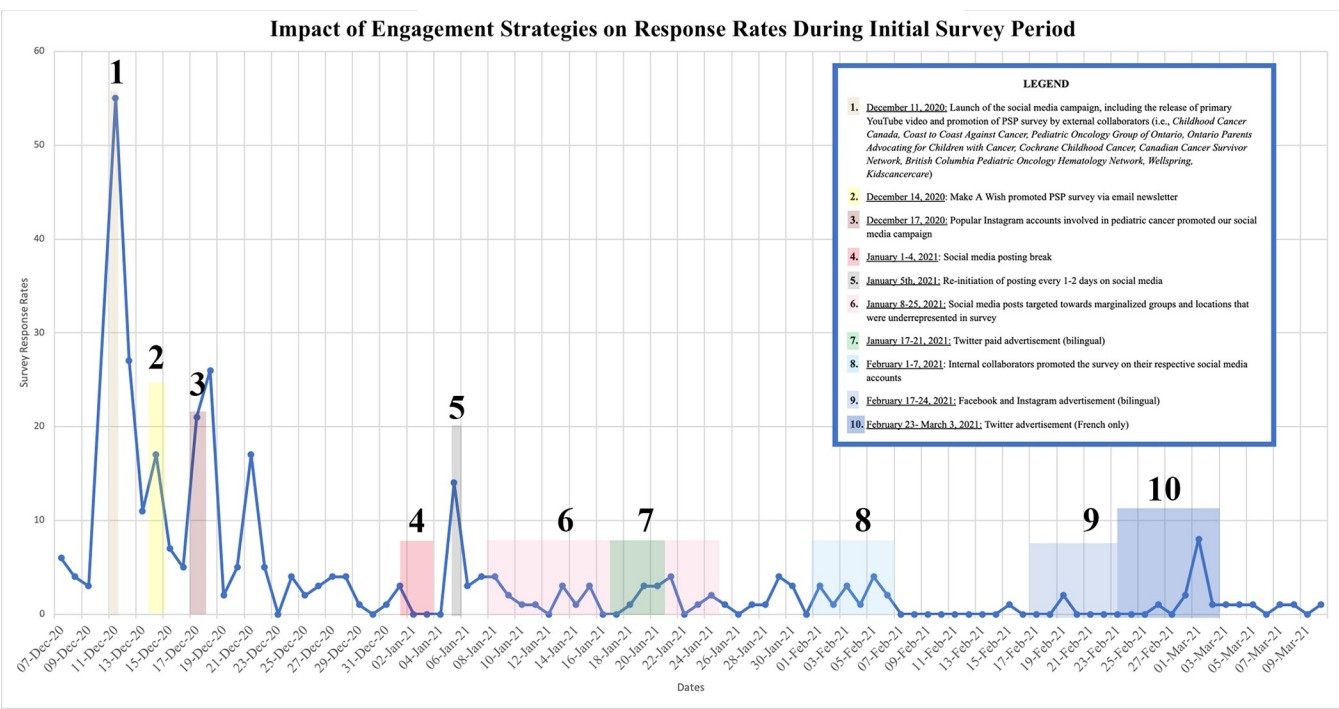

**Fig 4. Impact of engagement strategies on survey response rates during initial survey period.**

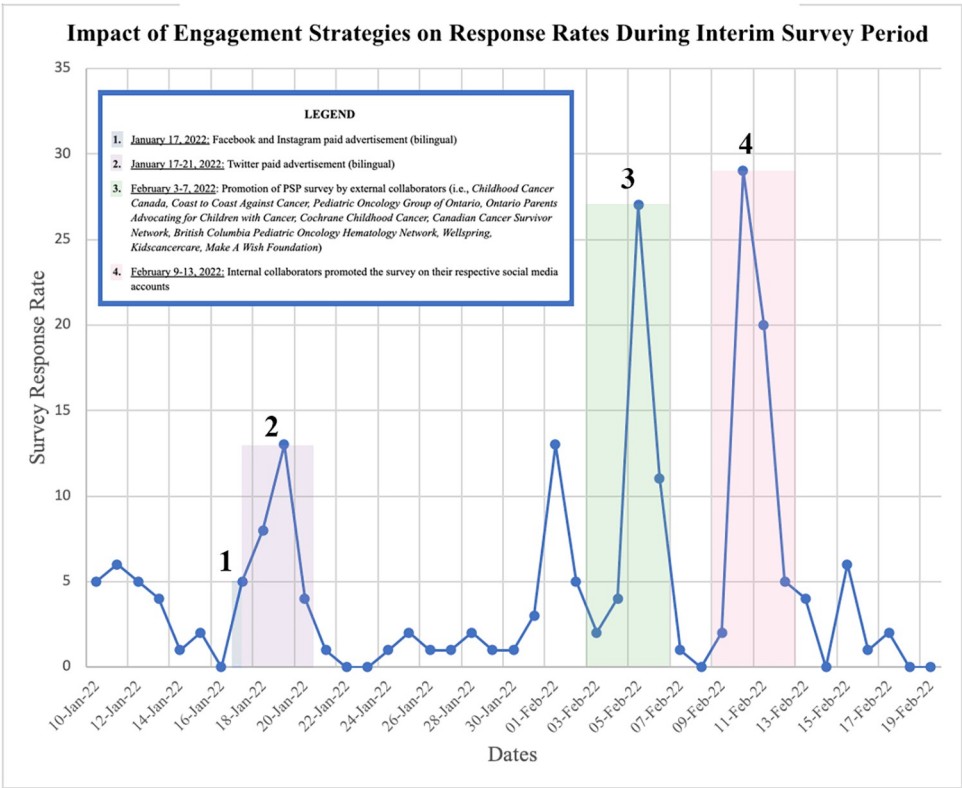

**Fig 5. Impact of engagement strategies on survey response rates during interim survey period.**

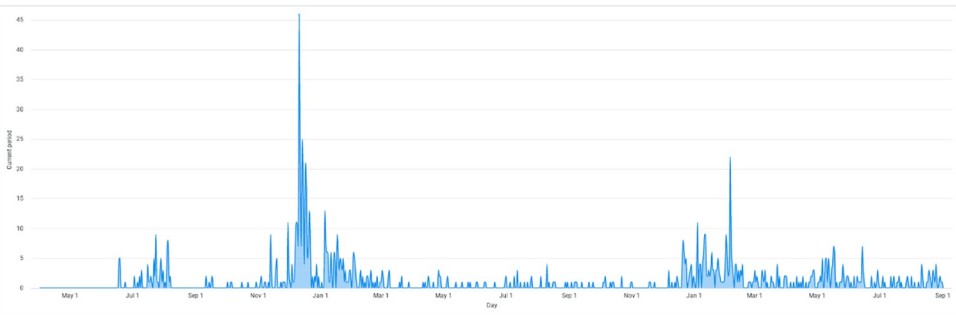

**Fig 6. Site session trend over social media campaign period.**

**Study website engagement.** The study website received 1029 site sessions (visits ended after 30 minutes of inactivity), with 53% of users (n = 547) accessing the site via desktop, 46% (n = 470) via a mobile phone and 1% (n = 12) via tablet. **Fig 6** depicts the site session trend over time. Peak site sessions occurred between December 2020 to March 2021, as well as January 2022 to March 2022—time periods that corresponded to the interval during which each survey was open.

The website garnered a total of 789 unique visitors, with most users accessing the website from Canada (n = 622, 78.8%), followed by the United States (n = 73, 9.3%). **Fig 7** depicts a map with the traffic by location. Most visitors were routed to the site from social media (n = 148, 32.7%), followed by direct referrals—or visitors typing our site address into their browser—(n = 146, 32.2%), organic searches—or visitors clinking on a search engine result—(n = 105, 23.1%), and other referrals—visitors clicking a link placed on another website that is not a search engine or social media platform (n = 54, 11.9%). **Table 4** provides more information on the specifics of the traffic source.

**Facebook, instagram and twitter-focused campaign engagement.** Over the course of the project, we made 152 posts on Facebook and 148 posts on Instagram. Our Facebook page was visited 426 times, while our Instagram profile had 1,893 visits. The reach of our Facebook page, or the number of individuals seeing any content from or about the page (including from others who interact with the page), was 28,641 people and our Instagram profile reach was 2,954 people (**Fig 8**). Considering activity related to our paid ads over Instagram and Facebook, our reach was 24,137 people and paid impressions, or number of times a post is viewed including by the same individual, was 40,721. Paid reach compared to campaign cost can be viewed in **Fig 9**. Our Twitter page garnered 452 followers. The greatest number of impressions during a 90-day period occurred in January to March 2021—a period within which both

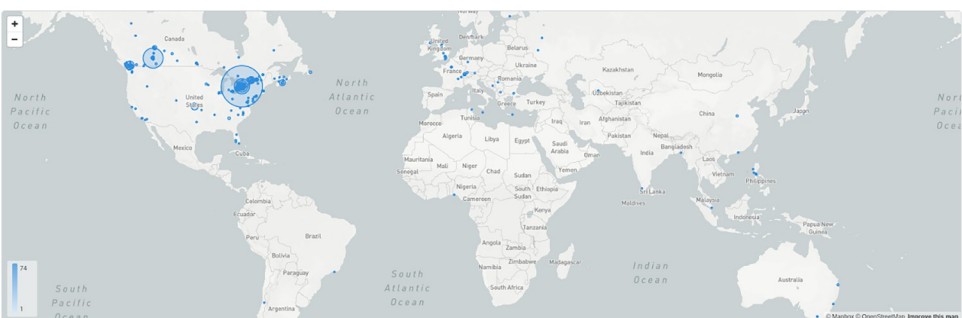

**Fig 7. Geographic distribution of website users (https://www.openstreetmap.org/#map=3/38.69/-72.42).**

**Table 4. Overview of website traffic sources.**

| Traffic category | Traffic source | Site sessions | Page views | Unique visitors |
|---|---|---|---|---|
| Direct | N/A | 186 | 203 | 146 |
| Organic search | Google | 161 | 199 | 94 |
| Social | Instagram | 59 | 66 | 51 |
| Social | Twitter | 56 | 66 | 42 |
| Social | Facebook | 47 | 58 | 46 |
| Referral | sickkids.ca | 26 | 26 | 25 |
| Referral | redcapexternal.research.sickkids.ca | 19 | 22 | 17 |
| Email | Email | 15 | 16 | 11 |
| Organic search | DuckDuckGo | 5 | 10 | 4 |
| Social | LinkedIn | 9 | 10 | 8 |
| Referral | Linktree | 5 | 5 | 5 |
| Organic search | Bing | 4 | 4 | 4 |
| Referral | pogon.convio.net | 2 | 3 | 2 |
| Referral | veeva.io | 2 | 2 | 2 |
| Organic search | Yahoo | 2 | 2 | 2 |
| Social | YouTube | 1 | 1 | 1 |
| Organic search | ecosia.org | 1 | 1 | 1 |
| Referral | outlook.live.com | 1 | 1 | 1 |
| Referral | us13.admin.mailchimp.com | 1 | 1 | 1 |
| Referral | com.linkedin.android | 1 | 1 | 1 |

organic posts and paid advertisements were implemented. **Fig 10** provides the number of impressions during 90-day time periods, the number of daily impressions, and details of the trend of impressions over time.

**Content and impact of individual social media posts.** The top reaching post over the course of our PSP was a French-language ad (**Fig 11**). This post explicitly asked persons with lived experience and healthcare professionals to participate in priority-setting, included related hashtags, and had an embedded link to the French language research prioritization survey. This advertised post reached 13,393 users, resulting in 200 clicks on the survey link, with approximately a cost of 0.69 Canadian dollars per click. The top unpaid post was an English-language Facebook post that reached 5,415 users and resulted in 47 link clicks. The majority of the top 10 posts across Facebook and Instagram were video-based (n = 9, 90%) as opposed to

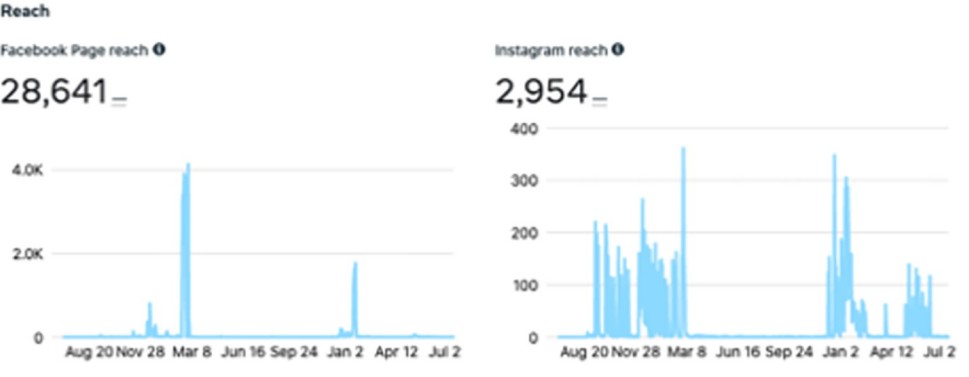

**Fig 8. Facebook and Instagram reach.**

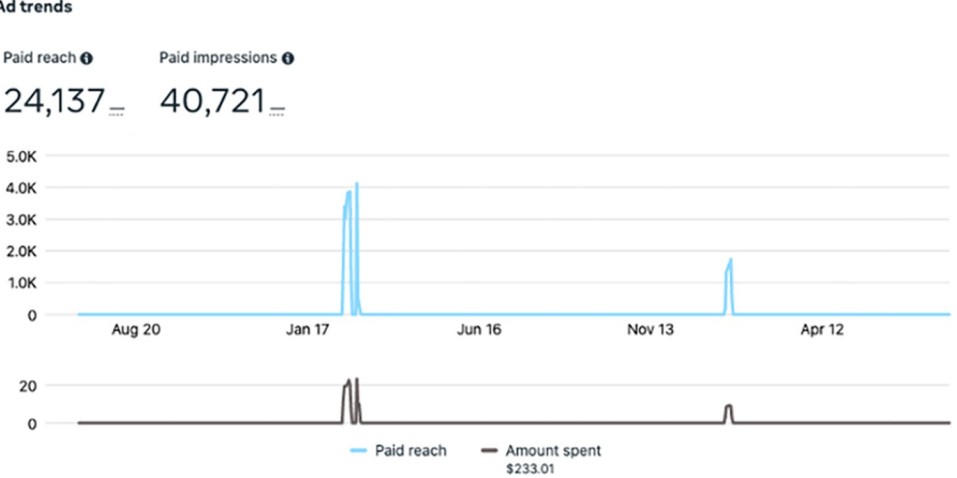

**Fig 9. Paid advertisement trends.**

including only static images. **Table 5** provides an overview of the top five posts, from both Facebook and Instagram, including both paid ads and unpaid posts.

## Discussion

### Principal findings

Social media represents a promising means to support health research, including the engagement in research priority-setting. Our research team used a multi-pronged strategy that leveraged social media and other online modalities to engage individuals with lived experience of childhood cancer and healthcare professionals in setting a Canadian research agenda Overall, our results demonstrate the utility of doing so and suggest such means may be useful to health researchers interested in engaging communities in PSP-style work.

Our campaigns on Facebook and Instagram had substantial success in reaching the Canadian pediatric cancer community with reaches of 28,641 and 2,954 people respectively, across both surveys. Further, our study website, which amassed over a thousand visits, saw most visitors being directed from our social media platforms. A previous pan-Canadian PSP integrated an in-person component where participants were informed of the study then provided with a device to record their responses with social media campaigning found most participants were recruited through social media (n = 337, 70.2%) [13]. Though this in-person recruitment approach was not available to our PSP due to the restrictions posed by the COVID-19 pandemic, this suggests that social media may be an effective and possibly resource lean means to build a large sample of participants.

Targeted paid advertising proved effective in increasing the reach of our social media campaign. Our posts with paid ad boosts reached more users than unpaid posts and each paid advertisement period was accompanied by a peak in the number of PSP surveys completed. The apparent success in engaging persons with lived experience and healthcare professionals by using paid ads reflect with previous research showing ads to reach many more users [14–16]. Based on our social media engagement analytics, video posts also tended to reach a broader audience than image-based posts [17]. Given this insight, our paid ads were mainly used with video content to further boost engagement.

In consultation with our steering group, we created social media content specifically targeting groups underrepresented in our PSP surveys to encourage their participation. These efforts

| Time Frame | Number of Impressions During Time Frame | Number of Daily Impressions | Type of Twitter Posts | Trend over time |
|---|---|---|---|---|
| Jul - Sep 2020 | 28.8K | 317 | Organic posts | Your Tweets earned 28.8K impressions over this 91 day period |
| Sep - Nov 2020 | 21.0K | 231 | Organic posts | Your Tweets earned 21.0K impressions over this 91 day period |
| Nov 2020 - Jan 2021 | 61.4K | 674 | Organic posts | Your Tweets earned 61.4K impressions over this 91 day period |
| Jan - Mar 2021 | 65.6K | 727 | Organic posts and paid advertisements | Your Tweets earned 65.5K impressions over this 90 day period |
| Apr - June 2021 | 5.4K | 60 | Organic posts | Your Tweets earned 5.4K impressions over this 90 day period |
| June - Sep 2021 | 3.6K | 41 | Organic posts | Your Tweets earned 3.6K impressions over this 90 day period |
| Sep - Dec 2021 | 5.1K | 57 | Organic posts | Your Tweets earned 5.1K impressions over this 90 day period |
| Dec 2021 - Mar 2022 | 43.7K | 480 | Organic post and paid advertisements | Your Tweets earned 43.7K impressions over this 91 day period |
| Mar - June 2022 | 12.5K | 137 | Organic posts | Your Tweets earned 12.5K impressions over this 91 day period |
| Jun - Aug 2022 | 1.6K | 18 | Organic posts | Your Tweets earned 1.6K impressions over this 66 day period |

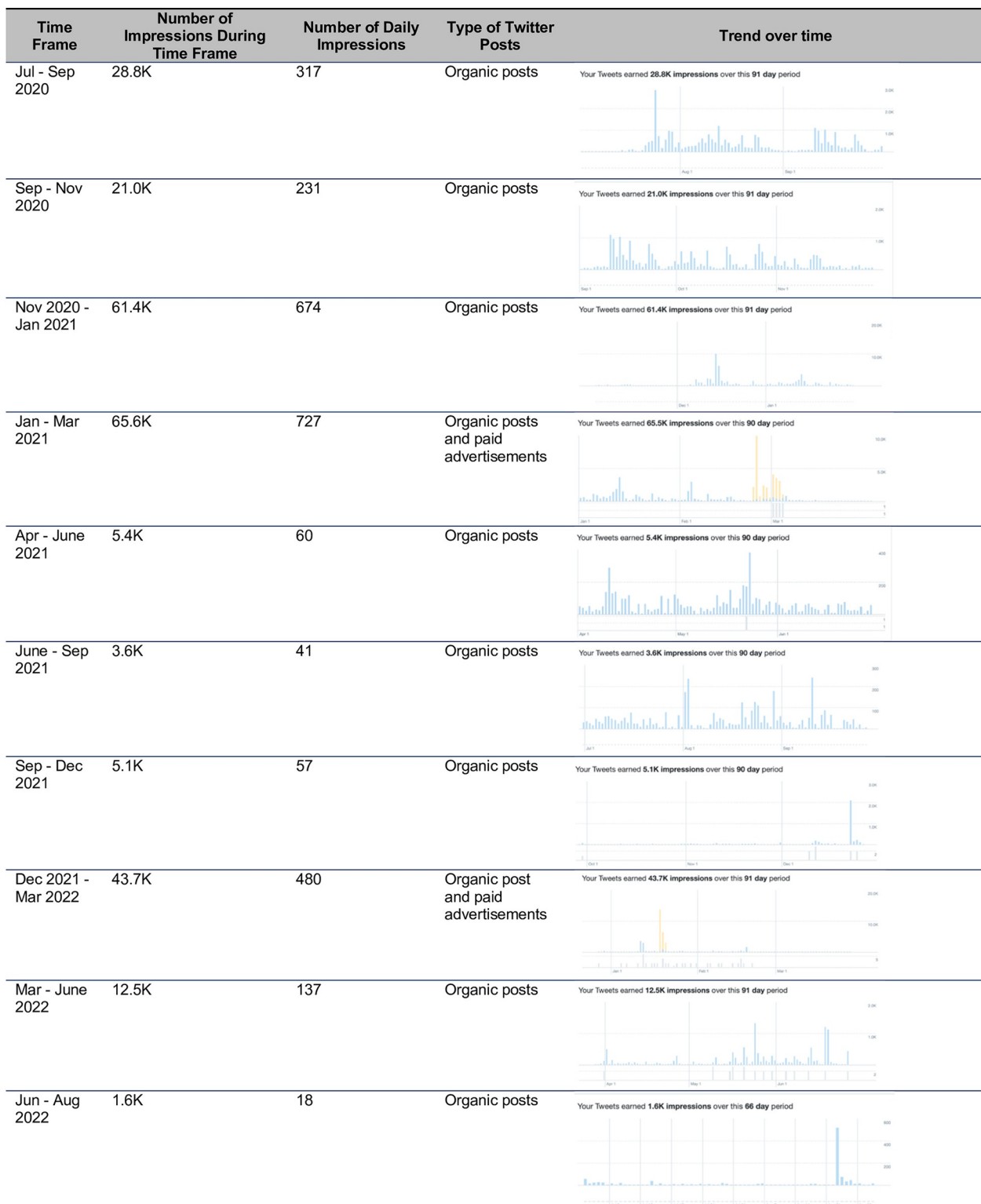

**Fig 10. Twitter analytics over 90-day time period.**

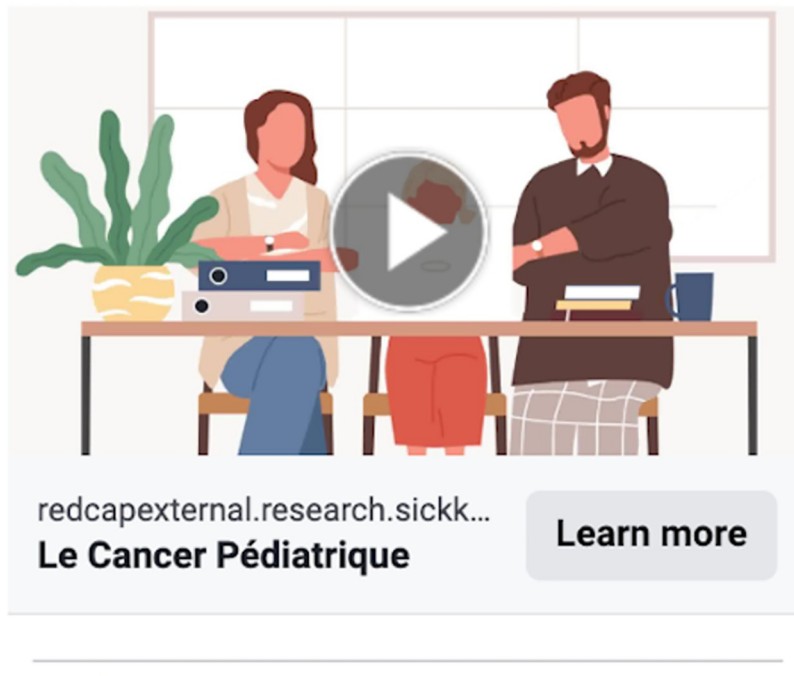

**Fig 11. Top reaching post during social media campaign duration.**

correlated temporally with an increased number of survey submissions from these groups. Additionally, to encourage pan-Canadian participation and representation from Francophone Canadians, our entire social media campaign was bilingual with both French and English language posts made. In these ways, we employed a tailored approach to reach specific populations on social media. In similar studies assessing the application of social media in health research, tailoring content to specific populations has been shown to enhance the engagement and user experience of these groups [18]. Reflecting the findings of our study, targeting content to particular groups has also been identified as an effective means to fill gaps in the reach of priority setting partnership survey promotions [19,20].

Our development of an external network of collaborative childhood cancer organizations and individuals was key to the success of our social media campaign. This strategy is embedded in the James Lind Alliance methodology and was an identified factor for success in

**Table 5. Top five posts from Facebook and Instagram.**

| Caption | Content type | Reach |
|---|---|---|
| Nous voulons entendre les patients, les survivants, les soignants, les parents endeuillés et les cliniciens! Quelles questions avez-vous sur le #cancerpédiatrique? Assurez vous que votre voix est entendue!<br><br>L'enquête ☞ cutt.ly/pedcancerpspfrench | Paid ad | 13393 |
| L'enquête de suivi tant attendue sur le #cancerpédiatrique est désormais LIVE! Selon vous, sur quoi la future #recherche sur le cancer pédiatrique devrait-elle se concentrer?<br><br>L'enquête ☞ bit.ly/pedcancerpspfrancais<br><br>Plus d'informations ☞ pedcancerpsp.ca 🎗<br><br>#childhoodcancer #pediatriccancer #pediatriccancerresearch #cancerresearch #childhoodcancerresearch #canceradvocate #kidsgetcancertoo #kidscancer #pediatriccancer #pediatriccancerawareness #smallbutmighty #research #pediatriccancerfoundation #teamworkmakesthedreamwork #joinus #helpushelpthem #makeadifference #Canada #nonprofit | French Facebook video post | 5415 |
| Are you a patient, survivor, caregiver, or clinician of #childhoodcancer? Help us make sure that future research focuses on what matters most to children with cancer by completing this survey: cutt.ly/pedcancerpsp CA 🎗 Survey is live!<br><br>More info ☞ pedcancerpsp.ca<br><br>Êtes-vous un patient, un survivant, un soignant ou un clinicien de #childhoodcancer? Aidez-nous à nous assurer que les recherches futures se concentrent sur ce qui compte le plus pour les enfants atteints de cancer en répondant à cette enquête: cutt.ly/pedcancerpsp CA 🎗<br><br>Plus d'infos ☞ pedcancerpsp.ca<br><br>#childhoodcancer #pediatriccancer #pediatriccancerresearch #cancerresearch #childhoodcancerresearch #canceradvocate #kidsgetcancertoo #kidscancer #pediatriccancer #pediatriccancerawareness #smallbutmighty #pediatriccancerfoundation #teamworkmakesthedreamwork #joinus #helpushelpthem #makeadifference #follow #share #nonprofit | English Facebook video post | 2101 |
| The long-awaited follow-up #childhoodcancer survey is now LIVE! What do you think future #research about childhood cancer should focus on?<br><br>Survey ☞ http://bit.ly/pedcancerpsp<br><br>More info ☞ http://pedcancerpsp.ca<br><br>We want to hear from all of Canada! Your voice matters 🎗 CA<br><br>#childhoodcancer #pediatriccancer #pediatriccancerresearch #cancerresearch #childhoodcancerresearch #canceradvocate #kidsgetcancertoo #kidscancer #pediatriccancer #pediatriccancerawareness #smallbutmighty #research #pediatriccancerfoundation #teamworkmakesthedreamwork #joinus #helpushelpthem #makeadifference #Canada #nonprofit | English Facebook video post | 1184 |
| Are you a patient, survivor, caregiver, bereaved parent or clinician impacted by #childhoodcancer? Help us make sure that future research focuses on what matters most to children with cancer by taking 10 minutes this holiday season to fill out this survey: cutt.ly/pedcancerpsp<br>Êtes-vous un patient, un survivant, un soignant, un parent endeuillé ou un clinicien touché par un cancer pédiatrique? Aidez-nous à nous assurer que les recherches futures se concentrent sur ce qui compte le plus pour les enfants atteints de cancer en prenant 10 minutes cette saison des fêtes pour répondre à ce sondage: cutt.ly/pedcancerpspfrench<br><br>#childhoodcancer #pediatriccancer #pediatriccancerresearch #cancerresearch #childhoodcancerresearch #canceradvocate #kidsgetcancertoo #kidscancer #pediatriccancer #pediatriccancerawareness #smallbutmighty #research #pediatriccancerfoundation #teamworkmakesthedreamwork #joinus #helpushelpthem #makeadifference #Canada #nonprofit | Bilingual Facebook picture post | 657 |

research priority-setting in our previous scoping review [7]. Known challenges in approaching members of the public to participate in health research complicate the recruitment process and are due in part to the inherent power dynamic that exists with healthcare providers [21]. Partnering with external organizations and communities that are known and trusted by a social media target audience can help to build campaign credibility and overcome historical challenges to PSP recruitment [22]. Although the possibility of spreading posted content widely and organically via social media exists, this is difficult without an established and well-connected social media presence [23]. Given that our project involved building and utilizing new social media channels with no prior following, we found partnerships with our external network and leveraging their established networks to be particularly valuable. Our partnerships with external organizations were bolstered by early contact, the provision of a template package of tailored graphics and messaging, brief reminders throughout the study period, and providing some degree of mutual benefit (e.g., study progress updates, providing a summary of study results). Table 6 provides an overview of the specific recommendations derived from our social media campaign.

**Strengths and limitations.** We observed a substantial gap between the number of visitors to our social media platforms and those that proceeded to open and complete the survey. This

**Table 6. Recommendations for future research-based social media campaign.**

| Recommendation Category | Specific Instructions | Applicable Platforms |
|---|---|---|
| Unifying Branding | Prior to the launch of social media campaigns, we recommend creating a branding package (i.e., standard fonts, colour palettes, and illustrations), as this facilitates unification of social media campaigns across various platforms | All (i.e., Facebook, Instagram, Twitter, LinkedIn, TikTok, Website, YouTube) |
| Accessibility | Graphics should meet minimum colour contrast requirements for web accessibility (https://webaim.org/resources/contrastchecker/) | All (i.e., Facebook, Instagram, Twitter, LinkedIn, TikTok, Website, YouTube) |
|  | Alternative text should be provided for all images and graphics | Website |
|  | Navigation should be simple and accessible on mobile and desktop devices | Website |
| Reflecting Target Audience | Text in posts should reflect target audience's language needs (e.g., all text was available in French and English, reflective of Canada's two official languages) | All (i.e., Facebook, Instagram, Twitter, LinkedIn, TikTok, Website, YouTube) |
|  | Stakeholders from the social media campaign's target audience should review the social media campaign prior to launch to ensure values of the target community are reflected in the campaign's content | All (i.e., Facebook, Instagram, Twitter, LinkedIn, TikTok, Website, YouTube) |
|  | Multiple platforms should be utilized, as different demographics may be associated with more frequent use of preferred social media platforms (e.g., researchers may frequently access LinkedIn, while pediatric patients may frequently access TikTok). | All (i.e., Facebook, Instagram, Twitter, LinkedIn, TikTok, Website, YouTube) |
|  | Texts and graphics should be intermittently tailored throughout the campaign to target specific under-represented groups (e.g., we intermittently posted targeted posts that included graphics with fathers taking care of a child with cancer, and text stating "Fathers of children with cancer, we want your voice heard in our survey") | Facebook, Instagram, Twitter, LinkedIn, TikTok, YouTube |
| Optimizing Campaign Reach | Complete website platform's requirements for SEO. The following checklist was provided by Wix:<br>1. Set the homepage title for search results<br>2. Add the homepage description for search results<br>3. Update the text on your homepage<br>4. Make your homepage visible in search results<br>5. Optimize your site for mobile devices<br>6. Connect your site to a custom domain<br>7. Connect your site to Google Search Console | Website |
|  | Implement paid advertisements via social media platforms to increase reach, and fill in gaps of under-represented populations | Facebook, Instagram, TikTok, LinkedIn, Twitter |
|  | Expand possible audience reach by collaborating with internal networks and external organizations by requesting they share social media campaign content; we recommend facilitating this collaboration by providing template packages. | Facebook, Instagram, TikTok, LinkedIn, Twitter, Email |
|  | Ensure hashtags are included in posts, as this assigns the designated post to the search results of the respective hashtag. | Facebook, Instagram, Twitter, LinkedIn, TikTok |
|  | Monitor analytics throughout the social media campaign to identify under-represented audiences and optimal posting times; adjust social media campaign accordingly. | Facebook, Instagram, Twitter, LinkedIn, TikTok, YouTube |

is a commonly cited limitation in PSPs, where the social media campaign reach is disproportionately greater than the actuate survey response rate [23]. Although we provided clear instructions on accessing and completing the survey in each social media post, elaborating on these details and improving capacity to navigate to the survey may have yielded a greater response rate. Nevertheless, despite these gaps between campaign engagement and our survey response rate, our PSP succeeded in securing similar numbers of participants to other pediatric illness PSPs and was successful in communicating and raising awareness about our project and its results within the pediatric cancer community and beyond. Another limitation of social media usage in priority-setting research is the uncertainty of who is being engaged with through the posts [24,25]. Social media–based methods may unexpectedly include or exclude the research priority perspectives of certain groups [7]. In our case, the team and steering committee ensured that social media posts were accompanied with a clear definition of the survey respondent eligibility criteria. Additionally, the survey was prefaced by a screening question

intended to filter out ineligible respondents. Despite these efforts, individuals that did not fit our PSP eligibility criteria were likely reached by our social media campaign and were still able to complete the survey. We also experienced initial difficulty engaging individuals from certain geographic regions. Particularly, it was challenging during the first to reach persons situated in Quebec. However, our use of posts and paid ads targeted towards this group and connections with Quebec-based external partners supported greater participation in the second survey. This reinforces the importance of supplementing social media posts with additional engagement network-based strategies. Finally, we recruited many more family members and healthcare professionals than children with cancer or young cancer survivors. To engage with youths, we attempted to use TikTok, one of the most utilized social media platforms among young people [26,27]. Still, this strategy had limited impact in terms of engagement. Despite our efforts to attract the younger age groups, we had limited data on the impact of these strategies. Recognizing that social media preferences among young people are constantly evolving, a continued understanding of forthcoming platforms among the younger population is warranted [28].

**Future research.** We recognize that little is known on the operation of social media algorithms. Further research is needed to understand how social media algorithms can influence recruitment to capture representative samples. More research is also needed to understand which social media strategies and dissemination techniques are likely to be successful for research prioritization efforts, with the understanding that these strategies and techniques are likely to change over time as new social media platforms and features become available.

The ethics requirements of traditional recruitment techniques are difficult to translate to research using social media given its potentially vast reach [23] and research in this area is also needed. Privacy risks also exist when social media-based recruitment methods are utilized. While our PSP only used social media to advertise, and recruitment and data collection occurred via a separate survey with data stored securely at our institution, further parameters to enhance participant privacy have been suggested [29]. Particular strategies include developing privacy notices for social media campaigns, creating disclaimers on the privacy risks of social media platforms, and disabling the comment feature, though consideration of the effectiveness of such strategies is needed [30].

## Conclusion

The engagement of people with lived healthcare experiences in research priority-setting is critical. We have presented our experience using social media to engage children with cancer, survivors, their family members, and healthcare providers in the development of a Canadian research agenda. Diversifying recruitment across multiple platforms increased response rates and improved the reach in an efficient manner. Utilizing paid ads, tailoring social media content to specific groups, and circulating promotional material to partnering external organizations were key strategies used to engage those with lived experience and healthcare professionals our PSP. Further investigation of social media algorithms and dissemination techniques is needed to understand how to increase representation among survey respondents. Consideration of the privacy implications of social media use for research is also needed. Ultimately, continued evaluation of novel tools to enable inclusive priority-setting may amplify the voices of those with lived experience as it pertains to the next scientific efforts.

## Author Contributions

**Conceptualization:** Kyobin Hwang, Surabhi Sivaratnam, Lindsay A. Jibb.

**Data curation:** Kyobin Hwang, Surabhi Sivaratnam.

**Formal analysis:** Kyobin Hwang, Surabhi Sivaratnam.

**Funding acquisition:** Lindsay A. Jibb.

**Investigation:** Kyobin Hwang, Surabhi Sivaratnam, Lindsay A. Jibb.

**Methodology:** Kyobin Hwang, Surabhi Sivaratnam, Lindsay A. Jibb.

**Project administration:** Elham Hashemi, Lindsay A. Jibb.

**Resources:** Lindsay A. Jibb.

**Supervision:** Elham Hashemi, Lindsay A. Jibb.

**Visualization:** Surabhi Sivaratnam.

**Writing – original draft:** Kyobin Hwang, Surabhi Sivaratnam.

**Writing – review & editing:** Kyobin Hwang, Surabhi Sivaratnam, Rita Azeredo, Elham Hashemi, Lindsay A. Jibb.

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
