## [Decision Letter · Decision Letter 0]

6 Jun 2023

PDIG-D-22-00360

Exploring the use of social media and online methods to engage knowledge users in creating research agendas: Lessons from a pediatric cancer research priority-setting partnership

PLOS Digital Health

Dear Dr. Jibb,

Thank you for submitting your manuscript to PLOS Digital Health. After careful consideration, we feel that it has merit but does not fully meet PLOS Digital Health's publication criteria as it currently stands. Therefore, we invite you to submit a revised version of the manuscript that addresses the points raised during the review process.

Please submit your revised manuscript within 60 days Aug 05 2023 11:59PM. If you will need more time than this to complete your revisions, please reply to this message or contact the journal office at digitalhealth@plos.org. Please include the following items when submitting your revised manuscript:

We look forward to receiving your revised manuscript.

Kind regards,

Yuan Lai, Ph.D.

Academic Editor

PLOS Digital Health

Journal Requirements:

a. State what role the funders took in the study. If the funders had no role in your study, please state: “The funders had no role in study design, data collection and analysis, decision to publish, or preparation of the manuscript.”

b. If any authors received a salary from any of your funders, please state which authors and which funders.

2. Please provide separate figure files in .tif or .eps format only and remove any figures embedded in your manuscript file. Please also ensure that all files are under our size limit of 10MB.

3. Fig 5: please (a) provide a direct link to the base layer of the map (i.e., the country or region border shape) and ensure this is also included in the figure legend; and (b) provide a link to the terms of use / license information for the base layer image or shapefile. We cannot publish proprietary or copyrighted maps (e.g. Google Maps, Mapquest) and the terms of use for your map base layer must be compatible with our CC-BY 4.0 license. 

Additional Editor Comments (if provided):

Reviewers' comments:

Reviewer's Responses to Questions

**Comments to the Author**

1. Does this manuscript meet PLOS Digital Health’s publication criteria? Is the manuscript technically sound, and do the data support the conclusions? The manuscript must describe methodologically and ethically rigorous research with conclusions that are appropriately drawn based on the data presented.

Reviewer #1: Partly

Reviewer #2: Partly

2. Has the statistical analysis been performed appropriately and rigorously?

Reviewer #1: N/A

Reviewer #2: Yes

3. Have the authors made all data underlying the findings in their manuscript fully available (please refer to the Data Availability Statement at the start of the manuscript PDF file)?

Reviewer #1: Yes

Reviewer #2: Yes

4. Is the manuscript presented in an intelligible fashion and written in standard English?

Reviewer #1: Yes

Reviewer #2: Yes

5. Review Comments to the Author

Reviewer #1: I am not an expert in this field of social media campaigns for priority setting. However, the paper was interesting to read and provided very detailed descriptions about this PSP. I found the discussion section about the lessons learned and a reflecton on the limitatons most interesting. Methods are informative for readers who want to do something similar. The results section is not very relevant in my opinion, because the results are merely descriptive. Without context, the number of hits or views is not that informative. I think there are some elements to add to the results. 

For example, what I missed in the results is information about the effort that was put into this campaign (in terms of hours and money for the paid advertisements), so that the claim of “efficiency” compared to other recruitment strategies can be judged better. I am convinced by the authors that their strategy to reach people had an effect and is certainly different from traditional approaches. But I cannot judge the “effectiveness” of the response, based on the data. Would it be interesting, for example, to place the results in a framework (possible an existing one) to summarize the working components of the strategy and its effectiveness? It is written in line 87 “methods to assess social media campaign effectiveness“ as outcome of your scoping review. What are these methods and which one has been applied to this study? How is effectiveness and efficiency defined? Especially since for this study not only the number of participants was relevant, but also the diversity of the group. 

Table 4: column Applicable platforms does not add a lot to your list of recommendations, or at least I do not understand what it intends to show. As you already point out, popular platforms come and go and also their features might change over time. Linking a specific recommendation to a specific platform might be outdated in the near future. Should you, however, decide to keep the specific platforms, I would suggest adding a small column for each platform and putting a symbol/x in the cells. Then it will be easier to see which recommendations apply to which platform. (currently, they are not in the same order in all cells)

For readability, I hope in the final paper all figures can be added as a supplemental data. 

Minor: in abstract the abbreviation PSP is not introduced

Last. I think an extra (positive) effect of this broad campaign is that many more people know about the topic, even though they do not participate in the prioritization effort. At least they know that this policy is using the ’voice of the people’, which might give them confidence that, ultimately, the research money will be well-spent. In that regard, the large number of people that do not respond to the social media posts should not be considered a waste. This aspect of using social media could be stressed as well.

Reviewer #2: I really enjoyed going through the paper. It is heartening to learn how the team has strategically leveraged social media to engage with participants. I, however, would request the authors to add more information into the manuscript along with some edits to consider. I have enlisted these below:

1. Please add a definition for 'knowledge user'. I struggled a bit trying to differentiate it from the word 'stakeholder' and 'social media user'. At places in the manuscript, I felt as if the 'knowledge user' and the 'social media user' words could be/ have been used interchangeably though there were some eligibility criteria set by the authors. Even these eligibility criteria have not been specified conspicuously in the paper.

2. The authors have generated the learnings in the process of 'experimenting' strategies with the social media campaigning. However, it is unclear how these learnings were consolidated. This should come up as a paragraph in the methods section. Was it an iterative consensus building effort by the team among themselves? Were these consolidated learnings validated in anyway with the help of domain experts e.g. social media marketing experts, campaign experts, etc?

3. As I read through the paper, I could appreciate the things that worked. But, what were the strategies that did not work and had to be phased out in the process. Of course, I could get that engaging with people from Quebec has been a challenge despite targeted efforts but did the team 'phase out' any particular effort?

4. Please provide the details of the team that managed the strategy e.g. number of members, their profiles, and if there were any prior experience that the team had in designing/ managing such campaigns.

5. I understand that some of the interventions e.g. placing paid ads were added to the strategy in due course. Could it be possible to include a flow diagram for a timeline so that the reader knows the total duration of the campaign and what was done when along with respective outputs? 

6. Please add the details of the each 'member category' (lines 72-73), Canadian Pediatric Cancer Community (line 75), full description paper (Line 79; if unavailable, please add a brief summary as a supplementary file), 'other online tools' (line 79-80), social media profiles that allowed 'business/ professional accounts' (line 99-100), 'key childhood cancer organizations (Line 110; how were these identified? Were these only from Canada?), 'social media users' (line 110; how were these identified?)

7. In line 131, the authors have identified that video posts were 'cost-effective'. How was this adjudged? Was any cost-effectiveness analysis done? What was the cost of the overall campaign and its specific elements?

8. Could the sentence in line 178 "social media campaigning efforts.... website" in the Results section be moved to the Methods section instead?

Once the authors add the above details, the paper will be very detailed to guide the reader interested in emulation.

Thank you!

6. PLOS authors have the option to publish the peer review history of their article (what does this mean?). If published, this will include your full peer review and any attached files.

**Do you want your identity to be public for this peer review?** For information about this choice, including consent withdrawal, please see our Privacy Policy.

Reviewer #1: No

Reviewer #2: Yes: Archisman Mohapatra

---

## [Decision Letter · Decision Letter 1]

6 Dec 2023

Exploring the use of social media and online methods to engage persons with lived experience and healthcare professionals in creating research agendas: lessons from a pediatric cancer research priority-setting partnership

PDIG-D-22-00360R1

Dear Dr. Jibb,

We are pleased to inform you that your manuscript 'Exploring the use of social media and online methods to engage persons with lived experience and healthcare professionals in creating research agendas: lessons from a pediatric cancer research priority-setting partnership' has been provisionally accepted for publication in PLOS Digital Health.

Best regards,

Yuan Lai, Ph.D.

Academic Editor

PLOS Digital Health

Reviewer Comments (if any, and for reference):

Reviewer's Responses to Questions

**Comments to the Author**

1. If the authors have adequately addressed your comments raised in a previous round of review and you feel that this manuscript is now acceptable for publication, you may indicate that here to bypass the “Comments to the Author” section, enter your conflict of interest statement in the “Confidential to Editor” section, and submit your "Accept" recommendation.

Reviewer #1: All comments have been addressed

2. Does this manuscript meet PLOS Digital Health’s publication criteria? Is the manuscript technically sound, and do the data support the conclusions? The manuscript must describe methodologically and ethically rigorous research with conclusions that are appropriately drawn based on the data presented.

Reviewer #1: (No Response)

3. Has the statistical analysis been performed appropriately and rigorously?

Reviewer #1: (No Response)

4. Have the authors made all data underlying the findings in their manuscript fully available (please refer to the Data Availability Statement at the start of the manuscript PDF file)?

Reviewer #1: (No Response)

5. Is the manuscript presented in an intelligible fashion and written in standard English?

Reviewer #1: (No Response)

6. Review Comments to the Author

Reviewer #1: (No Response)

7. PLOS authors have the option to publish the peer review history of their article (what does this mean?). If published, this will include your full peer review and any attached files.

**Do you want your identity to be public for this peer review?** For information about this choice, including consent withdrawal, please see our Privacy Policy.

Reviewer #1: None
